# Accuracy of Estimated Bioimpedance Parameters with Octapolar Segmental Bioimpedance Analysis

**DOI:** 10.3390/s22072681

**Published:** 2022-03-31

**Authors:** Fanglin Jiang, Saizhao Tang, Jin-Jong Eom, Keon-Hyoung Song, Hyeoijin Kim, Sochung Chung, Chul-Hyun Kim

**Affiliations:** 1National Traditional Sports Teaching and Research Section of Hunan Province, College of Physical Education, Hunan Normal University, Changsha 410012, China; jfl0108681@gmail.com; 2Department of Sports Medicine, Soonchunhyang University, Asan 31538, Korea; tangsaizhao@gmail.com; 3Department of Sport, Leisure & Recreation, Soonchunhyang University, Asan 31538, Korea; eomjjong@sch.ac.kr; 4Department of Pharmaceutical Engineering, Soonchunhyang University, Asan 31538, Korea; beophyen@sch.ac.kr; 5Department of Physical Education, Korean National University of Education, Cheongju-si 38173, Korea; aries408@knue.ac.kr; 6Department of Pediatrics, Konkuk University Medical Center, Konkuk University School of Medicine, Seoul 05030, Korea

**Keywords:** segmental bioimpedance, validity, estimated resistance, aging, body composition

## Abstract

The validity of the impedance parameters of the five body segments estimated using octapolar segmental bioelectrical impedance analysis (OS-BIA) has not been confirmed. This study aimed to verify the accuracy of the resistance (*R*), reactance (*Xc*), and phase angle of each five-body segment. The accuracy of the OS-BIA at 50 kHz was measured based on the direct tetrapolar segmental BIA. The differences in the estimated impedance parameters of the five body segments were compared to those measured from the OS-BIA in elderly men (*N* = 73) and women (*N* = 63). The estimated 50 kHz-*R* (Ω) was significantly higher than the measured 50 kHz-*R* in the right and left arms, and lower than the measured 50 kHz-*R* of the trunk, right leg, and left leg (all, *p* < 0.05). The estimated 50 kHz-phase angles in all the five body segments were significantly lower than the measured ones (all, *p* < 0.05). The findings suggest that the estimated impedance parameters, *R*, *Xc*, and phase angle of the trunk, were remarkedly underestimated, limiting the assessment of the physiological state of the organs in the body. Therefore, further intensive research is needed in the field of estimated segmental BIA in the future.

## 1. Introduction

Bioelectrical impedance analysis (BIA) is a portable, easy-to-use, and inexpensive method that is commonly used to assess total body water (TBW) and body composition [1,2]. It measures the electrical resistance (*R*) and impedance (*Z*) that are observed in the human body in response to electric current from surface electrodes. The *Z* parameters measured by BIA include resistance (*R*) and reactance (*Xc*). The phase angle (φ, PhA), which is directly calculated from *R* and *Xc* as arctangent (*Xc*/*R*) × (180/π), can be expressed for both the changes in the amount as well as the quality of the soft tissue mass, which is independent of conventional regression equations for estimating the body composition. Resistance is an impedance parameter generated by preventing the conduction of electric current into the body’s interstitial fluid (ISF) and TBW, including blood.

Reactance is an impedance parameter that is observed when conduction is delayed in the membranes of cells in the human body. When an AC flows through the human body, healthy cell membranes function as capacitors that store electrical energy, consequently causing a delay in the current’s flow. This lag in the current that penetrates the cell membranes and tissue interfaces creates the phase difference between the current and voltage, which is expressed as the PhA [3]. These impedance parameters are used as variables of prediction equations in cross-validation studies to develop predictive equations and are often used to indirectly assess TBW and body composition. However, body composition estimated using prediction equations was inferred from inaccurate assumptions and contained errors [3,4,5], and such limitations led to an attempt to evaluate and interpret physiological principles using raw impedance parameters, including the measured *R* and *Xc*, instead of indirectly estimating body composition from the impedance parameters [6,7].

To date, studies have shown that impedance parameters are highly correlated with the severity and prognosis of diseases such as human immunodeficiency virus infection, cancer, dialysis of patients with end-stage renal disease, malnutrition, and anorexia nervosa, suggesting that the raw values of impedance parameters may be used in clinical settings [8,9,10,11,12,13]. In previous studies, the phase angle (φ) was shown to be related to muscle mass [14]. A cross-sectional study reported a significant positive correlation between the phase angle and fat-free mass (FFM) and significant differences in resistance and reactance between well-trained bodybuilders and healthy control individuals [6,8]. In clinical studies, BIA parameters have been reported as prognostic factors in patients with heart failure [15,16,17]. These findings suggest the association between raw impedance parameters and health, disease, and physical fitness, thereby increasing the interest in the clinical application of these parameters [6,18,19,20,21].

Raw impedance parameters were measured using the traditional BIA technique with a tetrapolar (four-point electrode) segmental BIA(TS-BIA) at 50 kHz. Two current electrodes and two voltage electrodes are attached to the proximal and distal ends of a specific human body segment to measure the generated impedance parameters [1,22]. To perform segmental BIA, TS-BIA requires electrodes on the wrist, ankle, hip, and shoulder. In addition, subjects needed to remove part of their clothing during the measurement, which requires a long measurement time. Therefore, it is not suitable to perform field studies in populations. With the advancement of modern technology, the octapolar (eight-point electrode) segmental BIA (OS-BIA) was developed and commercialized recently [23,24]. There is growing interest in the development of diagnostic methods for sarcopenia, following the increase in the elderly population and prevalence of sarcopenia in developed countries. In addition, the European Working Group on Sarcopenia in Older People has recently recommended BIA to diagnose sarcopenia in the EWGSOP and EWGSOP2 sarcopenia guidelines, leading to increasing demand for the use of the OS-BIA [22,25,26,27].

The OS-BIA involves four current electrodes and four voltage electrodes on the ends of the extremities and analyzes the impedance parameters of arm-to-arm, right-arm-to-right-leg, left-arm-to-left-leg, and leg-to-leg circuits to calculate the impedance parameters of the right arm, right leg, trunk, left arm, and left leg based on the principle of the equipotential [26,27,28,29]. This new method effectively estimates the impedance parameters of the five body segments via one measurement with a simple procedure and minimal effort. However, the validity of the impedance parameters between the extremities and trunk, estimated by the OS-BIA, has not yet been confirmed. In other words, no study has compared the accuracy of the estimated impedance parameters, including *R*, *Xc*, and phase angle of the five body segments, in OS-BIA with the measured bioimpedance parameters of the body segments. The estimated impedance parameters in the body segments with the OS-BIA are used as predictor variables to develop a prediction equation of the body composition of each segment, as well as the whole human body. Because inaccurate values of the estimated impedance parameters must lead to errors in not only the impedance parameters of each segment but also the body composition of the segments and the total body composition [1,3,27], it is essential to verify the accuracy of the estimated impedance parameters of each body segment in the OS-BIA.

To the best of our knowledge, no previous studies have verified the accuracy and validity of the estimated segmental impedance parameters obtained using the OS-BIA technique. Several previous studies have assessed the accuracy of the whole-body impedance parameters obtained using the OS-BIA technique and compared these to those obtained using the TS-BIA technique and reported significant differences in the impedance parameters of the whole body in healthy adult women [30], hemodialysis patients [31], bodybuilders [32], the elderly [33], and children [34,35]. Therefore, this study aimed to verify the accuracy of the impedance parameters of each body segment as estimated using the OS-BIA method based on the direct TS-BIA method in the elderly who required regular BIA assessments.

## 2. Materials and Methods

### 2.1. Participants

The participants of the current study comprised a total of 136 elderly people over the age of 65, including 73 men and 63 women. The participants were recruited through local newspapers and social media advertisements, and those who were registered at the public health center of borough offices and voluntarily wished to participate were included. The study was approved by the ethical committee of the Korean National Sport University (No.1263-201903-HR-010-02) and performed in accordance with the Declaration of Helsinki. The exclusion criteria were as follows: contraindications for BIA (such as neurological disease, musculoskeletal disorders, heart failure, renal disease, edema, cachexia, open wounds, rash, cardiac pacemakers, and metal implants), individuals who had been hospitalized within the last 3 months, or individuals who had a history of limb amputation. Prior to the BIA, all participants fasted for 4–6 h. Body weight was measured in units of 0.5 kg (CAS DB-1, Seoul, South Korea) in light clothes, and the height was measured in 1 mm units (SECA 274, Hamburg, Germany).

### 2.2. Measurement of the Direct Segmental Impedance Parameters

Quantum Desktop RJL-101 (RJL Systems, Clinton Twp, MI, USA), a 4-point electrode (tetrapolar) real-time BIA device, was used as the criterion measure to assess the accuracy of the estimated segmental impedance parameters from the OS-BIA. Participants were instructed to refrain from strenuous exercise, alcohol, and caffeine intake for the 24 h before the experiment, finish their last meal at least 4 h before the measurement, and empty their bladder within 30 min before the measurement. Participants adopted a supine position with limbs slightly spread apart from the body (Figure 1a). Adhesive gel electrodes were placed at defined anatomical sites on the anterior surface of the trunk and the dorsal surfaces of the hands, wrists, ankles, and feet on the right and left side as explained in the following. The measurements of the right and left arm were taken by attaching one pair of electrodes on the hand and wrist and the other pair of electrodes on the anterior surface of the shoulder. For the shoulder, the superior edge of the current electrode was attached on the inferior border of two-thirds of the clavicle and the voltage electrode 5 cm inferior to this point. For the hand and wrist, the proximal edge of the detecting (voltage) electrode was attached to form an imaginary line bisecting the styloid process of the ulna and the proximal edge of the source (current) electrode on an imaginary line bisecting the metacarpophalangeal joint of the middle finger [36,37]. Trunk impedance was measured by placing one pair of electrodes on the anterior surface of the shoulder and the other pair of electrodes on the surface of the lingual crest on the anterior midline of the proximal thigh on the right and left sides. For the right and left shoulders, the superior edge of the source electrode on each side was attached over the inferior border of lateral two-thirds of the clavicle and the detecting electrode 5 cm inferior to this point. For the lingual crest, the detecting electrode was attached to the anterior superior iliac spine and the detecting electrode 5 cm superior to the source electrode [36,37]. Leg impedance of each side was measured by placing one pair of electrodes on the foot and ankle and placing the other pair of electrodes on the surface of the lingual crest on the anterior midline of the proximal thigh. For the iliac crest of each side, the detecting electrode was attached to the ASIS (anterior superior iliac spine), and for the iliac crest of each side, the detecting electrode was attached to the ASIS and the source electrode 5 cm superior to the detecting electrode. For the ankle and foot of each side, the proximal edge of the ankle detecting electrode was attached to form an imaginary line bisecting the medial malleolus, and the distal edge of the foot source electrode was placed to form an imaginary line through the metatarsophalangeal joints of the second and third toes [36,37].

### 2.3. Assessments of Estimated Octapolar Segmental Impedance Parameters

An octapolar segmental BIA device (InBody S10, InBody, Seoul, Korea) was used to estimate the impedance parameters of the five segments (right arm, right leg, trunk, left arm, and left leg) using the basic principle of the equipotential technique [28,29]. The participants were asked to lie in the supine position with both arms 15° from the body and legs 15° apart from each other for OS-BIA. The electrodes were attached according to the manufacturer’s recommendations, and their locations were the same as those for the hands, wrists, ankles, and feet during the TS-BIA.

For the hand and wrist of each hand, the proximal edge of the detecting electrode was attached to form an imaginary line bisecting the styloid process of the ulna and the distal edge of the current electrode on an imaginary line bisecting the metacarpophalangeal joint of the middle finger. For the ankle and foot, the proximal edge of the ankle electrode was attached to form an imaginary line bisecting the medial malleolus, and the distal edge of the current electrode was placed to form an imaginary line through the metatarsophalangeal joints of the second and third toes [36,37] (Figure 1b). A single examiner measured all the impedance parameters during the octapolar and tetrapolar segmental BIA. To assess the intratester reliability, impedance parameters obtained from OS-BIA and TS-BIA were evaluated in fifteen subjects on one day at two consecutive times. The electrodes used for OS- and TS-BIA were repositioned before each measurement. Each evaluation was performed by the same tester. Intraclass correlation coefficient (ICC) of the examiner for the repeatability was conducted for the single tester. The intraclass correlation coefficients of the examiner for the repeatability were 0.961 and 0.955, respectively.

### 2.4. Fat Mass and Fat-Free Mass from Dual-Energy X-ray Absorptiometry

A whole-body DXA Prodigy Advance scanner (GE Lunar, Madison, WI, USA) was used to measure each participant’s total fat mass and fat-free mass. The DXA instrument was calibrated daily using the spine phantom provided by the manufacturer. For standardization purposes of the scans, the files from the original DXA machine were transferred to iDXA Software, version 4.0.2. Whole-body scans were manually analyzed for the manufacture-defined regions of interest (ROI) following the standard analysis protocol of the *GE Lunar User Manual*.

### 2.5. Data Processing and Statistical Analysis

Descriptive statistics were analyzed for resistance, reactance, and phase angle, and the normality of all variables was confirmed. A paired t-test was conducted for the mean differences between the estimated and measured data. Agreement between the two methods was assessed using a linear regression analysis. The intraclass correlation coefficient (ICC) was used to evaluate the consistency between the two methods. IBM SPSS Statistics (version 23.0; IBM Corporation, Armonk, NY, USA) was used for data analysis, and a *p*-value less than 0.05 was considered statistically significant.

## 3. Results

### Characteristics of the Study Population

One hundred and thirty-six older adults participated in this study. The total sample was divided into groups of 73 men and 63 women to validate the estimated bioimpedance parameters, including *R*, *Xc*, and the phase angle. The general characteristics of the two study groups are summarized in Table 1. There were no between-group differences in age, BMI, and *Xc*. Men were taller, weighed more, had lower FM, FMI, %BF, *Z*, and *R*, and higher FFM, FFMI, and phase angles than women (all *p* < 0.05).

The accuracies of the estimated *R*, *Xc*, and *PhA* of the segmental bodies in elderly men and women are presented in Table 2 and Figure 2. The estimated *R* values in men and women were significantly higher than the measured *R* values in the right and left arms and lower than those in the right and left legs (all *p* < 0.001). In addition, *R* of the trunk in men and women was significantly underestimated compared to that measured (*p* < 0.001). The ratio of *R* of the estimated to measured impedance parameters was high for the arms and low for the trunk and right and left legs. For *Xc* in men, there were no differences between the right and left arms. However, the estimated *Xc* of the right and left arms were higher than that of the right and left arms in women. The estimated *Xc* of men and women was significantly underestimated in the right and left legs (all *p* < 0.001). In addition, *Xc* of the trunk in men and women was significantly underestimated compared to that measured using the four-electrode bioimpedance technique (*p* < 0.001). Therefore, the trunk and right and left legs had a low ratio of reactance. Meanwhile, the accuracy of the estimated phase angle (*PhA*) in older men and women was significantly underestimated in all five segments (all *p* < 0.001).

## 4. Discussion

This study aimed to verify the accuracy of the impedance parameters of the body segments estimated using the OS-BIA technique and compare their values with those obtained using the direct TS-BIA technique in the elderly who required regular BIA assessments. The differences in the estimated and directly measured impedance parameters of the segmental bodies in elderly men and women were compared. In our study, the estimated segmental resistances of the upper and lower extremities were overvalued and undervalued, respectively, when compared to those obtained from the direct TS-BIA technique. The estimated segmental resistance of the trunk was severely undervalued, with a low value. Similar trends were observed for estimated segmental reactance. The values of the upper and lower extremities were overvalued and undervalued, respectively, and that of the trunk was severely undervalued. As a result, the phase angle, which indicates the standardized resistance and reactance, measured using eight-electrode BIA, was significantly lower than that obtained from direct four-electrode BIA in both elderly men and women. In particular, the estimated reactance of the trunk was only 0.22 times the measured reactance. These results suggest that the impedance parameters, *R*, *Xc*, and phase angle, of the lower extremity and trunk were highly undervalued and needed to be calibrated. The results of the current study show that the trunk and lower extremities should be assessed carefully when developing a predictive equation for body composition and appendicular skeletal muscle because the underestimated segmental impedances and resistance must be reduced the *Z* index (=Ht^2^/Z), which can overvalue the TBW (total body water = ρZ index), FFM (=TBW/0.7372), and sarcopenia (or muscle mass). Additionally, the underestimated phase angle of all the five segments should represent underestimation of cell body mass, low integrity, and distorted homeostasis. The results of this study are consistent with those of several previous studies, showing that the original estimated impedance parameters must be carefully interpreted because they were either undervalued or overvalued. Several reasons may account for such differences in the segmental impedance parameters between the two techniques. First, differences in the positions of the current and voltage electrodes may lead to different impedance parameters. According to the cylindrical model proposed by Cornish et al. (1999), inaccurate placement of the electrodes on the ends of the arms or legs and a 1 cm shift in the position of the electrodes may cause an error of 2% in the measured impedance parameters [38]. Therefore, the position of the electrodes may affect the measurement of the whole body or the segmental impedance parameters. In a recent study, Shiffman (2013) demonstrated that different electrode characteristics significantly affect the resistance [39]. A slight shift in the position of the electrodes can have significant effects on the resistance of the whole body as the current flows through the disproportionate ends of the extremities [40]. However, we used a well-defined region for the electrode attachment location, and a single researcher reliably performed all assessments. Therefore, the bias in the electrode position was considered insignificant in this study. Second, the path of the current for the TS-BIA technique is clearly defined as being composed of just a single circuit from the right hand to the right forearm, trunk, right thigh, right leg, and right foot; however, the path of the current for the OS-BIA technique has not been clearly established, even though the principle of equipotential is quite clear [1,7,26,27,28,29]. Studies have demonstrated that there are six current circuits (arm-to-arm, right-arm–left-leg, right-arm–right-leg, left-arm–left-leg, right-arm–right-leg, foot-to-foot) for the equipotential-based OS-BIA technique [28,29]; however, these circuits are not clearly defined, especially for the determination of reactance and the phase angle. Importantly, studies have reported additional body current circuits, such as right-arm–right-leg, left-leg–right-leg circuits, and impedance parameters in these current circuits can overlap and lead to distorted results in impedance values [1,37,41]. This overlap of the current circuits could be eliminated by positioning the arm to be adducted 90 degrees at the shoulder joint, so that the voltage between the measured and unmeasured volage electrode is equipotential. Alternatively, the overlap can be removed by calculating the degree of overlap using an algorithm validated through several experiments. BIA parameters have been used in different clinical settings and as disease prognostic and mortality predictors. Therefore, it is necessary to conduct research to remove the overlapping errors caused by the double-track circuit in the future. Accurate raw impedance parameters of the trunk without masking errors will be of great help in the study of internal organs for physiological and pathophysiological metabolisms in the future.

The limitation of this study is that the TS-BIA requires electrodes on the wrist, ankle, hip, and shoulder. In addition, the subjects had to remove a part of their clothing during the measurement, which required a longer measurement, which required a longer measurement time. Meanwhile, this new method, OS-BIA, effectively estimates the impedance parameters of the five body segments with a simple procedure and minimal effort. However, the validity of the impedance parameters between the extremities and trunk, estimated using the OS-BIA technique, has not yet been confirmed. Therefore, the estimated impedance parameters of OS-BIA were validated by the TS-BIA based on 136 healthy elderly people aged ≥65 years. Given the sample size of the population under study, the specific age group and health status of the subjects cannot represent the adults, youth, and patients, which may affect the generalizability of our results.

## 5. Conclusions

In conclusion, the OS-BIA impedance parameters of the lower extremities and trunk were markedly undervalued, leading to overestimation of the Z index, FFM, and sarcopenia. In addition, the phase angles of all five segments were undervalued, which distorted cell membrane integrity and homeostasis, limiting the evaluation of the physiological state of body organs. Therefore, more intensive research on OS-BIA is needed in the future.

## Figures and Tables

**Figure 1 sensors-22-02681-f001:**
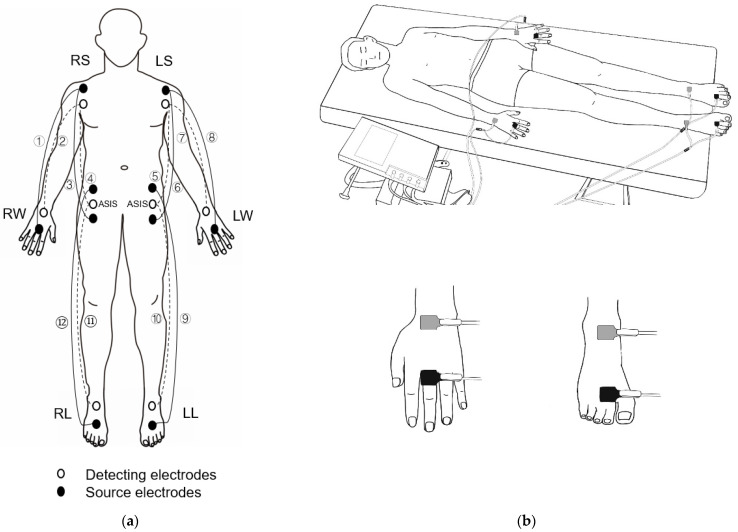
The posture for assessment and locations of the electrodes. (**a**) The direct tetrapolar segmental BIA and (**b**) octapolar segmental BIA. (① a pair of source electrodes on the right arm, ② a pair of detecting electrodes on the right arm, ③ a pair of source electrodes on the right trunk, ④ a pair of detecting electrodes on the right trunk, ⑤ a pair of detecting electrodes on the left trunk, ⑥ a pair of source electrodes on the left trunk, ⑦ a pair of detecting electrodes of the left arm ⑧ a pair of source electrodes of the left arm, ⑨ a pair of source electrodes on the left lower limb, ⑩ a pair of detecting electrodes of the left lower limb, ⑪ a pair of detecting electrodes on the right limb, and ⑫ a pair of source electrodes of the right limb. The trunk impedance value with the TS-BIA is mean to left and right.)

**Figure 2 sensors-22-02681-f002:**
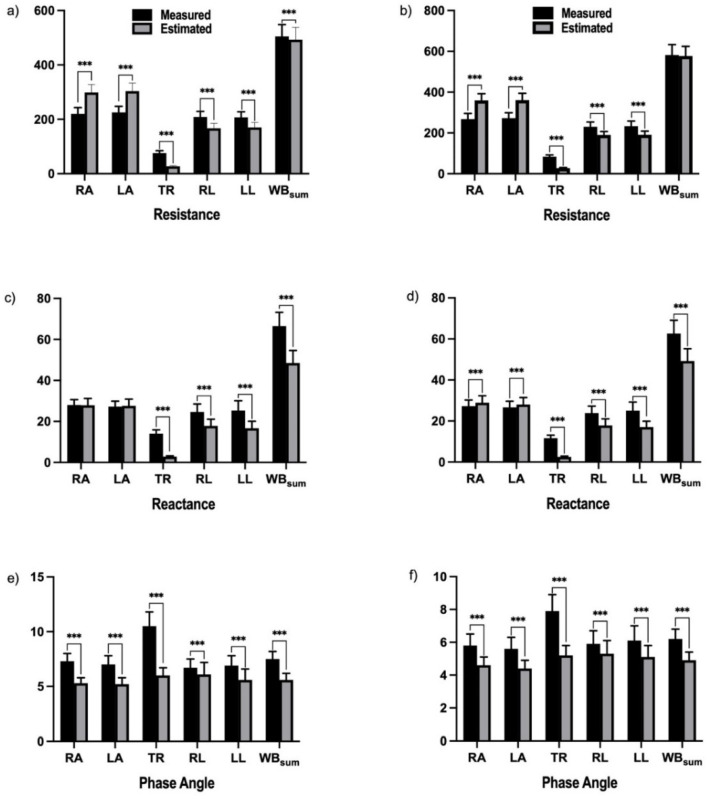
Comparison of the estimated impedance parameters to the measured impedance parameters with respect to resistance (*R*), mean reactance (*Xc*), and mean phase angle (*PhA*) measured at 50 kHz frequency. (**a**,**c**,**e**) correspond to *R*, *Xc*, and *PhA* values in males; (**b**,**d**,**f**) correspond to *R*, *Xc*, and *PhA* values in females. WB_sum_: Whole body as the sum of the segments for right arm, trunk, and right leg. *** = difference between the estimated and measured values at *p* < 0.001.

**Table 1 sensors-22-02681-t001:** General characteristics of elderly participants.

	Men (*n* = 73)	Women (*n* = 63)	Total (*n* = 136)
Age (year)	76.5 ± 4.1	75.1 ± 4.0	75.9 ± 4.1
Weight (kg)	65.4 ± 7.1	54.1 ± 6.1 ***	60.2 ± 8.7
Height (cm)	166.5 ± 4.8	152.2 ± 5.0 ***	159.9 ± 8.6
BMI (kg/m^2^)	23.6 ± 2.2	23.4 ± 2.2	23.5 ± 2.2
FFM (kg)	49.9 ± 4.3	36.9 ± 3.6 ***	43.9 ± 7.6
FM (kg)	15.5 ± 5.1	17.3 ± 3.9 *	16.3 ± 4.7
FFMI (kg/m^2^)	18.0 ± 1.1	15.9 ± 1.1 ***	17.0 ± 1.5
FMI (kg/m^2^)	5.6 ± 1.8	7.4 ± 1.6 ***	6.5 ± 2.0
%BF	23.3 ± 6.2	31.6 ± 4.8 ***	27.2 ± 6.9
*Z*_measured_ (Ω)	458 ± 41.5	537 ± 47.9 ***	495 ± 59.3
*R*_measured_ (Ω)	456 ± 41.5	535 ± 47.9 ***	492 ± 59.4
*Xc*_measured_ (Ω)	47.7 ± 5.3	48.4 ± 5.4	48.0 ± 5.4
*PhA*_measured_ (°)	6.0 ± 0.6	5.2 ± 0.6 ***	5.6 ± 0.7

Values are mean ± SD; %BF, percentage of body fat; BMI, body mass index; FFM, fat-free mass; FFMI, fat-free mass index; FM, fat mass; FMI, fat mass index; *PhA*, phase angle; *R*, resistance; *Xc*, reactance; *Z*, impedance; *Z*, *R*, *Xc,* and *PhA* are the whole-body electrical impedance parameters from 4-point electrode bioelectrical impedance analysis, and whole body as assessed by right-arm-to-right-foot 4-electrode BIA; FM and FFM were assessed by DXA; * = significantly different from females at *p* < 0.05; *** = significantly different from females at *p* < 0.001.

**Table 2 sensors-22-02681-t002:** Comparison of the estimated impedance parameters to the measured impedance parameters.

	Male (*n* = 73)			Female (*n* = 63)		
	Measured ^†^	Estimated ^‡^	Ratio	Measured	Estimated	Ratio
*Resistance* (*R*, Ω)						
RA	220 ± 22.8	299 ± 29.5 ***	1.36 ± 0.06	268 ± 28.3	360 ± 33.0 ***	1.36 ± 0.06
LA	225 ± 22.5	304 ± 29.1 ***	1.35 ± 0.06	272 ± 26.4	361 ± 33.1 ***	1.33 ± 0.06
TR	76 ± 8.6	27 ± 2.7 ***	0.36 ± 0.03	84 ± 8.7	27 ± 2.7 ***	0.33 ± 0.03
RL	209 ± 20.7	167 ± 18.1 ***	0.80 ± 0.07	230 ± 23.0	190 ± 16.4 ***	0.83 ± 0.05
LL	207 ± 20.5	170 ± 18.8 ***	0.82 ± 0.07	233 ± 25.0	191 ± 17.8 ***	0.82 ± 0.05
WB_estimate_	505 ± 43.8	493 ± 45.5 ***	0.98 ± 0.05	582 ± 51.8	577 ± 46.7	0.99 ± 0.05
WB_measure_	456 ± 41.5	493 ± 45.5 ***	1.08 ± 0.04	535 ± 47.9	577 ± 46.7 ***	1.08 ± 0.04
*Reactance* (*Xc*, Ω)						
RA	28.0 ± 2.6	27.9 ± 3.3	1.00 ± 0.07	27.2 ± 3.0	8.9 ± 3.4 ***	0.33 ± 0.08
LA	27.2 ± 2.7	27.6 ± 3.3	1.02 ± 0.08	26.6 ± 3.0	8.0 ± 3.4 ***	0.30 ± 0.08
TR	14.0 ± 1.9	2.8 ± 0.4 ***	0.21 ± 0.03	11.6 ± 1.5	2.5 ± 0.4 ***	0.22 ± 0.03
RL	24.6 ± 3.9	17.8 ± 3.3 ***	0.73 ± 0.13	23.8 ± 3.4	17.8 ± 3.2 ***	0.75 ± 0.08
LL	25.3 ± 4.8	16.7 ± 3.4 ***	0.66 ± 0.10	25.0 ± 4.2	17.0 ± 2.9 ***	0.69 ± 0.10
WB_estimate_	66.5 ± 6.7	48.5 ± 6.1 ***	0.73 ± 0.07	62.6 ± 6.5	49.5 ± 6.0 ***	0.79 ± 0.06
WB_measure_	47.7 ± 5.3	48.5 ± 6.1	1.02 ± 0.07	48.4 ± 5.4	49.2 ± 6.0 *	1.02 ± 0.06
*Phage Angle* (°)						
RA	7.3 ± 0.7	5.3 ± 0.5 ***	0.74 ± 0.05	5.8 ± 0.7	4.6 ± 0.5 ***	0.79 ± 0.06
LA	7.0 ± 0.8	5.2 ± 0.6 ***	0.76 ± 0.06	5.6 ± 0.7	4.4 ± 0.5 ***	0.80 ± 0.06
TR	10.5 ± 1.3	6.0 ± 0.7 ***	0.58 ± 0.09	7.9 ± 1.0	5.2 ± 0.6 ***	0.66 ± 0.09
RL	6.7 ± 0.8	6.1 ± 1.1 ***	0.91 ± 0.17	5.9 ± 0.8	5.3 ± 0.8 ***	0.90 ± 0.10
LL	6.9 ± 0.9	5.6 ± 1.0 ***	0.81 ± 0.11	6.1 ± 0.9	5.1 ± 0.7 ***	0.84 ± 0.11
WB_estimate_	7.5 ± 0.7	5.6 ± 0.6 ***	0.75 ± 0.07	6.2 ± 0.6	4.9 ± 0.5 ***	0.79 ± 0.06
WB_measure_	6.0 ± 0.6	5.6 ± 0.6 ***	0.94 ± 0.07	5.2 ± 0.6	4.9 ± 0.5 ***	0.94 ± 0.06

Values are mean ± SD; ^†^ = measured 4-electrode BIA parameters; ^‡^ = estimated 8-electrode BIA parameters; RA: right arm, LA: left arm, TR: trunk, RL: right leg, LL: left leg; WB_estimate_: Whole body as the sum of the segmental for right arm, trunk, and right leg. WB_measure_: Whole body as assessed by right-arm-to-right-foot 4-electrode BIA; * = Difference between the estimated and measured values at *p* < 0.05; *** = difference between the estimated and measured values at *p* < 0.001.

## Data Availability

Not applicable.

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
