# Peer review of "Accuracy of Estimated Bioimpedance Parameters with Octapolar Segmental Bioimpedance Analysis"

_sensors, 2022, doi:10.3390/s22072681_

Round 1

Reviewer 1 Report

  • Please revise the English of the paper due to typos.
  • The authors should better define the rationale according to which the use of octapolar BIA would overcome the use of tetrapolar. The use of electrodes and the need for use the health budget for them could reduce the use of octapolar BIA. Please discuss.
  • the reproducibility of the methods should be better described. What about c-statistics? What is the incremental value of octapolar as compared to tetrapolatr BIA?
  • the clinical implication of this analysis should be also discussed. FOr example, Scicchitano P and his group described the impact of BIA in heart failure setting (Scicchitano P et al. Biomedicines. 2021 Sep 24;9(10):1307; Scicchitano P et al. Heart Lung. 2020 Nov-Dec;49(6):724-728; and Scicchitano P, Massari F.Biomark Med. 2020 Feb;14(2):81-85). Please discuss such a point.
  • the limitations of each technique should be described.

Author Response

  • Please revise the English of the paper due to typos. 

    Response: We got the paper proofread by a professional proofreading company.

  • The authors should better define the rationale according to which the use of octapolar BIA would overcome the use of tetrapolar. The use of electrodes and the need for use the health budget for them could reduce the use of octapolar BIA. Please discuss.
    Response: We thank the reviewer for this pertinent comment. Accordingly, we described in Introduction that OS-BIA is clinically more useful and practical than TS-BIA. 

    After revision: 

    In clinical studies, BIA parameters have been reported as prognostic factors in patients with heart failure [15-17]. These findings suggest the association between raw impedance parameters and health, disease, and physical fitness, thereby increasing the clinical application of these parameters (See line 70-74)

    To perform segmental BIA, TS-BIA requires electrodes on the wrist, ankle, hip, and shoulder. In addition, subjects needed to remove part of their clothing during the measurement, which required a long measurement time. Therefore, it is not suitable to perform field studies in populations. (See line 78-81)

  • the reproducibility of the methods should be better described. What about c-statistics? What is the incremental value of octapolar as compared to tetrapolatr BIA?

    Response: We added the details of the reproducibility testing procedure and ICC in the method.

    After revision: To assess the intrarater reliability, impedance parameters obtained from OS-BIA and TS-BIA were evaluated in fifteen subjects on one day two consecutive times. The electrodes used for OS- and TS-BIA were repositioned before each measurement. Each evaluation was performed by the same tester. Intraclass correlation coefficient (ICC) of the examiner for the repeatability was conducted for the single tester. The intraclass correlation coefficients of the examiner for the repeatability were 0.961 and 0.955, respectively. (See line 211-216)
  • the clinical implication of this analysis should be also discussed. FOr example, Scicchitano P and his group described the impact of BIA in heart failure setting (Scicchitano P et al. Biomedicines. 2021 Sep 24;9(10):1307; Scicchitano P et al. Heart Lung. 2020 Nov-Dec;49(6):724-728; and Scicchitano P, Massari F.Biomark Med. 2020 Feb;14(2):81-85). Please discuss such a point.

    Response: We thank the reviewer for this pertinent comment. We described the clinical significance and added the references in the revised manuscript. 

    After revision: In clinical studies, BIA parameters have been reported as prognostic factors in patients with heart failure. These findings suggest the association between raw impedance parameters and health, disease, and physical fitness, thereby increasing the interest in the clinical application of these parameters. (See line 70-74)

  • the limitations of each technique should be described.

    Response: We thank the reviewer for this pertinent suggestion. Accordingly, the limitations of the two techniques were described in the revised manuscript.

    After revision: TS-BIA requires electrodes on the wrist, ankle, hip, and shoulder. In addition, subjects needed to remove part of their clothing during the measurement, which required a longer measurement time. ~ Given the sample size of the population under study, the specific age group and health status of the subjects cannot represent the adults, youth, and patients, which may affect the generalizability of our results. (See line 346-356)

Reviewer 2 Report

The impedance parameters of five body segments were estimated using octapolar segmental bioelectrical impedance analysis . This study aims to verify the accuracy of resistance, reactance, and phase angle, of five parts of the human body, left and right hands and legs and trunk, by measuring their bioelectrical impedance at 50Hz. From comparison between estimated and measured parameters, for men and women, are found differences of values of impedances .

The paper concludes that the findings show that the estimated impedance parameters of the trunk are significantly lower, and limit the assessment of the physiological state of the organs in the body.

Is this reviewer opinion that the article will be significantly improved following the issues below:

The paper must be carefully revised for English language: style, syntax, grammar, vocabulary, order of words, etc. Because of poor language, some parts of article are difficult to follow.

In Abstract, the many numerical values are redundant and confusing: the abstract must summarize the main ideas only.

In this paper, the phase angle must be clearly defined. Which exactly is the role of phase angle ? Why are not enough the resistance and impedance?

The role and need for using the Bivariate linear regression (section3.2.)  in interpretation of measurements must be clarified. Is this helping to improve the conclusions? Otherwise remove section 3.2.

In Section 4. Discussion, the third paragraph is more a review of literature than a discussion of findings of this paper. Authors should consider to restructure the paper and move this paragraph to Section 1. Introduction.

The conclusions are very  generic: the statement that the estimated values are lower than the measured ones, is obvious. Practically, all measured values are smaller or higher than the estimated.

Such conclusions are not solid, because do not bring any actions that should be taken to correct the measuring errors. There are no proposals for remedies of the measuring errors. There are no procedures how to diminish the errors, or how  to limit them, or even how to cancel such differences.

Theauthors should consider to add one section with procedures forminimozing or cancelling measurement errors, to improve the  assessment of the physiological state of the organs in the human body .

 References without doi are difficult to retrieve.

Author Response

  • The impedance parameters of five body segments were estimated using octapolar segmental bioelectrical impedance analysis . This study aims to verify the accuracy of resistance, reactance, and phase angle, of five parts of the human body, left and right hands and legs and trunk, by measuring their bioelectrical impedance at 50Hz. From comparison between estimated and measured parameters, for men and women, are found differences of values of impedances .
  • The paper concludes that the findings show that the estimated impedance parameters of the trunk are significantly lower, and limit the assessment of the physiological state of the organs in the body.

    Is this reviewer opinion that the article will be significantly improved following the issues below:

  • The paper must be carefully revised for English language: style, syntax, grammar, vocabulary, order of words, etc. Because of poor language, some parts of article are difficult to follow.

    Response: We got the paper proofread by a professional proofreading company.

    After  revision: Proofed by a native English scientist (See manuscript.)

  • In Abstract, the many numerical values are redundant and confusing: the abstract must summarize the main ideas only

    Response: We thank the reviewer for this pertinent comment. Accordingly, we revised Abstract to enhance its clarity.

    After revision: Revised (See Abstract.)

  • In this paper, the phase angle must be clearly defined. Which exactly is the role of phase angle? Why are not enough the resistance and impedance?

    Response: We thank the reviewer for this pertinent suggestion. The phase angle is described in detail in Introduction. In addition, regarding 'no enough R and Z', we have added a detailed description in the discussion section. 

    After revison: The phase angle (φ, PhA), which is directly calculated from R and Xc as arc-tangent (Xc/R) ´ (180/p), can be ~ , which is expressed as the PhA’ (See line 41-63)

  • The role and need for using the Bivariate linear regression (section3.2.)  in interpretation of measurements must be clarified. Is this helping to improve the conclusions? Otherwise remove section 3.2

    Response: We deleted Table 3 because another reviewer and you have pointed out that bivariate linear regression is not necessary. 

    After revision: Deleted 

  • In Section 4. Discussion, the third paragraph is more a review of literature than a discussion of findings of this paper. Authors should consider to restructure the paper and move this paragraph to Section 1. Introduction.

    response: We thank the reviewer for this pertinent suggestions. The third paragraph in Discussion is abbreviated and described in the introduction

    After revision: “This new method effectively estimates the impedance parameters of the five body segments through one measurement ~ by the OS-BIA, has not been confirmed. (See line 93-96.)

  • The conclusions are very generic: the statement that the estimated values are lower than the measured ones, is obvious. Practically, all measured values are smaller or higher than the estimated

    Response: I am deeply thankful for you. I think that the quality of this reviesed manuscript has been improved because of your critical comment. 

    After revision: 
    the OS-BIA impedance parameters of the lower extremities and trunk were markedly underestimated, leading to overestimation of the Z index, FFM, and sarcopenia. In addition, the phase angle of all five segments were underestimated, which distorted cell membrane integrity and homeostasis, limiting the evaluation of the physiological state of body organs. Therefore, more intensive research on OS-BIA is needed in the future .

  • Such conclusions are not solid, because do not bring any actions that should be taken to correct the measuring errors. There are no proposals for remedies of the measuring errors. There are no procedures how to diminish the errors, or how  to limit them, or even how to cancel such differences.

    Response: Rewritten as directed

    After revision: However, the path of current of equipotential is quite clear [1,7,26-29,41] ~ overlap using an algorithm validated through several experiments. (see 340~344)

  • The authors should consider to add one section with procedures forminimozing or cancelling measurement errors, to improve the assessment of the physiological state of the organs in the human body.

    Response: Rewritten as directed

    After revision: However, the path of current of equipotential is quite clear [1,7,26-29,41] ~ overlap using an algorithm validated through several experiments. (see 340~344)

  • References without doi are difficult to retrieve.

    Response: We added doi to the references.

    After revision: See references.

Reviewer 3 Report

The manuscript verifies the accuracy of OS-BIA applied in impedance parameters investigations. The authors find the estimated impedance parameters were significantly underestimated. Reasons for the differences between OS-BIA and TS-BIA were also proposed. This is a systematic work.

I have two questions.

First, how do readers get the estimated data of Table 2 from the data of Table 1?

Second, why bivariate linear regression is needed, and what is the bivariate linear regression equation. These should be clear in the manuscript.

Author Response

  • The manuscript verifies the accuracy of OS-BIA applied in impedance parameters investigations. The authors find the estimated impedance parameters were significantly underestimated. Reasons for the differences between OS-BIA and TS-BIA were also proposed. This is a systematic work.
  • I have two questions.
  • First, how do readers get the estimated data of Table 2 from the data of Table 1?

    Response: We have studied a cohort study on sarcopenia in Korean population. This study, now submitted for review, was the first step toward a cohort study. Furthermore, as we were compiling this paper, we felt that the methods of FM and FFM did not need to be explained to obtain the body composition using DXA (dual-energy X-ray absorptiometry). However, because you enquired how we got our body composition, we explained how to measure FM and FFM using DXA in the methods section of the revised manuscript. In addition, the method for obtaining Zmeasured, Rmeasured, Xcmeasured, and PhAmeasured is described in the comment section of Table 1.

    After revision: 

    2.5 Fat mass and Fat-free mass from Dual-Energy X-Ray Absorptiometry ~ GE Lunar User Manual. (See line 227 ~ 235)

    After: , and whole body as assessed by right arm-to-right foot 4-electrode BIA~ female at p < .05; (See Table 1)

  • Second, why bivariate linear regression is needed, and what is the bivariate linear regression equation. These should be clear in the manuscript.

    Response: We thank the reviewer for this pertinent comment. We have deleted Table 3 because you and another reviewer pointed out that bivariate linear regression is not necessary. 

    After revision: Deleted 

Round 2

Reviewer 1 Report

The authors well addressed my previous comments. The paper improved very much.

Author Response

The authors well addressed my previous comments. The paper improved very much.

Response: Thank you very much. We could improve our manuscript by your comments. Thank you.

Reviewer 2 Report

Taking into consideration the authors' changes in the revised manuscript, i can recommend this paper to advance for publication. 

A carefull spell check will be welcome, such as the meaning of word "underestimate".

Author Response

Taking into consideration the authors' changes in the revised manuscript, i can recommend this paper to advance for publication. 

A carefull spell check will be welcome, such as the meaning of word "underestimate".

Response: We checked and correct some word including "underestimate"

after revision: arc-tangent => arctangent (see line 42)
                        intrarater -> intratester (see line 184)
                       overestimate / underestimate -> overvalued / undervalued
                       (see line 251 ~ 256, 267, 271, 273, 276)

I am deeply thankful for your comment. We could improve our manuscript according to your comments. Thanks again.